# **CLEO:** The Numerical Methods of a New Superdroplet Model including a Droplet Breakup Algorithm

Clara J.A. Bayley<sup>1, 2</sup>, Ann Kristin Naumann<sup>1, 3, 4</sup>, Florian Poydenot<sup>3</sup>, Raphaela Vogel<sup>3</sup>, Bjorn Stevens<sup>1</sup>, and Shin-Ichiro Shima<sup>5</sup>

**Correspondence:** Clara J.A. Bayley (clara.bayley@mpimet.mpg.de)

Abstract. The numerical methods of conventional Eulerian models have obscured our fundamental understanding of cloud microphysics and introduced artificial uncertainties. In contrast, the Super-Droplet Model (SDM) provides a transparent link between model and theory, and remedies the numerical artifacts that hindered decades of cloud modelling. In light of its numerous advantages we've created a novel SDM for warm-cloud microphysics called CLEO, with the goal of making it feasible to run Large-Eddy Simulations (LES) with SDM in domains large enough to resolve shallow mesoscale cloud organisation O(100 km). Here we document the microphysics grounding CLEO and how it is translated into numerical methods, with the intention of assisting the physical interpretation of future LES and comparison with observations. We highlight subtle but important points where we differ from existing SDMs: in how we model the ventilation effect on evaporation, and how we account for uncertainty in our knowledge of droplet collisions. As well as modelling collision-coalescence we propose a low-cost extension to the original SDM algorithm which adds both collisional rebound and breakup. We demonstrate CLEO's capabilities with known test-cases for condensation/evaporation, collisions between droplets, and droplet motion, including an integrated test using the 1-D Kinematic Driver framework. CLEO can therefore now be used stand-alone, one-way coupled for piggybacking, or two-way coupled for LES, as a fully-functioning SDM capable of representing all the main microphysical processes driving warm-clouds.

#### 15 1 Introduction

Substantial differences exist between conventional Eulerian models for cloud microphysics. Excessive degrees of freedom within one- and two-moment bulk schemes, the least expensive of existing microphysics models, allow for varying definitions of condensate categories, as well as the parametrisations for how they interact with one another, and their assumed size distributions (e.g. Khain et al., 2015). Increasing in sophistication, even the most conventional spectral bin schemes still vary in how they define condensate categories and their parametrisations. They also introduce differences in numerical implementations,

<sup>&</sup>lt;sup>1</sup>Max-Planck-Institut für Meteorologie, Hamburg, Germany

<sup>&</sup>lt;sup>2</sup>International Max Planck Research School on Earth System Modelling, Hamburg, Germany

<sup>&</sup>lt;sup>3</sup>Meteorologisches Institut, Universität Hamburg, Hamburg, Germany

<sup>&</sup>lt;sup>4</sup>Ludwig-Maximilians-Universität München, Munich, Germany

<sup>&</sup>lt;sup>5</sup>Graduate School of Information Science, University of Hyogo, Kobe, Japan

for example to tackle numerical diffusion (Morrison et al., 2020), such that even if two bin schemes solve the same equations they could calculate different results. These differences matter for more than just microphysics.

The differences between conventional Eulerian models have a wide-range of macrophysical ramifications. In global storm resolving models (e.g. Miyakawa et al., 2014; Bretherton, 2015), only changing fall velocity parameters in one- and two-moment bulk schemes can substantially alter the relationship between moisture and convection, which has consequences for the Madden–Julian oscillation and the radiation balance of the entire atmosphere (Suematsu et al., 2021; Takasuka et al., 2024; Naumann et al., 2025). Varying definitions of condensate size distributions are also problematic because they alter shallow-cloud fraction and albedo, even in non-precipitating clouds, and their underlying assumptions are inconsistent with even the most basic kinematic processes (Igel and van den Heever, 2017a, b). When Large-Eddy Simulations (LES) and observations are compared, differing precipitation patterns are frequently attributed to differences in microphysics schemes, and bin schemes are just as disparate as bulk ones (vanZanten et al., 2011; King et al., 2015; Schulz and Stevens, 2023). Indeed there is extensive literature showing the many ways in which microphysics can appreciably effect larger-scale dynamics (e.g. Barnes and Garstang, 1982; Hagos et al., 2018; Jian et al., 2021; Gasparini et al., 2023), and thus as long as there are differences between microphysics schemes we can expect there to be differences in macrophysical outcomes. Differences inevitably arise from gaps in our knowledge of cloud microphysics, however, the differences highlighted above are not caused by knowledge gaps, but rather by "model uncertainty", uncertainty inherent to the way that conventional Eulerian models represent what we already understand.

The Super-Droplet Model (SDM; Shima et al., 2009) eradicates the biggest sources of conventional model uncertainty. SDM is a Lagrangian particle model for cloud microphysics where the condensate population is modelled by "superdroplets". Superdroplets have the same properties as real condensates as well as an extra one, the multiplicity, which states how many real condensate particles that superdroplet represents. SDM therefore makes no assumptions about condensate categories or size distributions and it does not suffer from same intrinsic problems as bin schemes, such as numerical diffusion and the "curse of dimensionality" (Grabowski et al., 2019). In the same way the accuracy of the condensate size distributions is bounded by the number of moments in a bulk scheme, and the number of bins in a bin scheme, it is the number of superdroplets which limit the accuracy of SDM. However SDM has an important convergence property. As the number of superdroplets increases, a simulation tends towards what we expect from explicit individual particle simulations, and the mean of many simulations approaches the solution to the Smoluchowski equation (Smoluchowski, 1916). This is in stark contrast to bulk and bin schemes, where increasing refinement does not converge to a generic solution because the field-based description of particles has conceptual and numerical flaws (Grabowski et al., 2019; Morrison et al., 2020). Moreover, the link between our fundamental understanding of microphysical processes and their representation in SDM is clear and direct, meaning that variations between SDMs hold a strong relation to differences in our understanding of the underlying physics.

We have created a novel computational implementation of SDM, CLEO, for a simplistic representation of warm-cloud microphysics well-suited to high-performance computers. In the companion to this paper we described the fundamental computational design of our SDM and its computational performance (Bayley et al., 2025a). In this paper we document CLEO's numerical methods for warm-cloud microphysics and demonstrate their behaviour in known test-cases. Since the link between

the numerical methods and fundamental theory is so clear in SDM, by carefully describing our numerical methods we assist the physical interpretation of studies using CLEO and comparisons with other microphysics models.

CLEO models all the major microphysical processes of warm-clouds: condensation/evaporation, collisions between droplets and droplet motion. Most of the methods are already found in the literature, but in addition to documenting which of those CLEO adopts, we highlight particular novelties of our approaches and explain how the flexibility we incorporated reflects the uncertainties in our current understanding of warm-cloud microphysics. More specifically, we clarify the methods of Shima et al. (2009) and Matsushima et al. (2023) for modelling condensation/evaporation and extend them to include ventilation effects. For collision between droplets, CLEO facilitates various definitions for the collision kernel and the outcome of collisions between droplets. The breakup of raindrops as a consequence of collisions is included in de Jong et al. (2022) and Bringi et al. (2020), but is neglected by the majority of SDMs (and some bulk and bin microphysics schemes), despite the fact it could be an important influence on the rate of evaporation in downdraughts (Stevens and Seifert, 2008; Morrison and Milbrandt, 2011; Morrison et al., 2012; Seifert and Heus, 2013; Planche et al., 2019). CLEO therefore offers an extended version of the Shima et al. (2009) collision-coalescence algorithm to permit the study of rebound and breakup.

In this paper the  $i^{\text{th}}$  superdroplet is defined by its spatial coordinates,  $x_i$ , its multiplicity,  $\xi_i$ , and its attributes,  $a_i$ , wherein  $R_i$  and  $M_{s,i}$  are its radius and its mass of solute (aerosol), respectively. The total mass of each superdroplet is given by

$$M_{T,i} = M_{s,i} \left(1 - \frac{\rho_l}{\rho_{s,i}}\right) + \frac{4}{3} \pi \rho_l R_i^3, \tag{1}$$

to account for the solute's volume whilst neglecting its change due to dissolution, where  $\rho_l$  and  $\rho_{s,i}$  are the densities of liquid water and the solute, respectively.  $a_i'$  symbolises  $x_i$  as well as all the superdroplet's intensive attributes, for example  $\rho_{s,i}$ .

CLEO's numerical methods for microphysics and superdroplet motion are found in Sections 2 to 4, and their validations are found in Section 5. Section 2 describes how CLEO models condensation/evaporation, including ventilation effects and an adaptive sub-time-stepping algorithm. Section 3 explains how we have extended the collision-coalescence algorithm of Shima et al. (2009) to include a framework for collisional breakup and rebound, and Section 4 explains how CLEO determines superdroplet motion.

# 2 Condensation/Evaporation

75

We use Köhler theory to describe droplet growth due to evaporation/condensation, but also account for the enhancement of evaporation of moving droplets by including a ventilation factor. The change of the  $i^{th}$  superdroplet's radius,  $R_i$ , is therefore

$$R_i \frac{dR_i}{dt} = f_v \frac{(S-1) - \frac{a}{R_i} + \frac{b}{R_i^3}}{F_k + F_d},$$
(2)

where  $f_{\rm v}$  is the ventilation factor; S is the ambient saturation ratio of the grid-box;  $F_{\rm k}$  and  $F_{\rm d}$  are, respectively, the heat conductivity and vapour diffusion factors;  $\frac{a}{R_i}$  represents the effect of curvature on the saturation at a droplet's surface; and  $\frac{b}{R_i^3}$  accounts for the reduction in water vapour pressure due to the presence of solute. We use the same formulations for  $F_{\rm k}$ ,  $F_{\rm d}$ , a and b as in Shima et al. (2009) (Köhler, 1936; Rogers R.R., 1989). Supersaturation fluctuations caused by sub-grid-scale turbulence are not taken into account.

**Figure 1.** The ventilation factor we use in CLEO compared to the experimental data from Kinzer and Gunn (1951), and Pruppacher and Rasmussen (1979), and compared to the fit to the data from Pruppacher and Klett (1978) at  $T = (288.15 \pm 15.00)$  K in a standard atmosphere.

We obtain the ventilation factor,  $f_v$ , by fitting the curve

$$f_{\rm v}(R) = 1 + \left(\frac{1}{\alpha_1 R^{\beta_1}} + \frac{1}{\alpha_2 R^{\beta_2}}\right)^{-1} \tag{3}$$

to the curve from Pruppacher and Klett (1978), such that  $\alpha_1 = 6.954 \times 10^7 \,\mathrm{m}^{-\beta_1}$ ,  $\alpha_2 = 1.069 \times 10^3 \,\mathrm{m}^{-\beta_2}$ ,  $\beta_1 = 1.963$ ,  $\beta_2 = 0.702$ . This gives similar values as Pruppacher and Klett (1978) (given the terminal velocity parametrisation from Pruppacher and Klett, 1978) but makes the size dependence more explicit whilst neglecting the small dependence of  $f_v$  on pressure variations. Additionally we constrain  $f_v \leq 20.0$  to reflect the fact that droplets with a radius larger than approximately 3 mm have the same terminal velocity. As shown in Figure 1,  $f_v$  is negligible for droplets with radii less than about  $0.1 \,\mathrm{mm}$ , but can increase the evaporation rate of large droplets by an order of magnitude.

Careful consideration of the numerical methods for integrating equation 2 is required because this ordinary differential equation is stiff. We use an implicit Euler method with a Newton-Raphson root-finding algorithm similar to Shima et al. (2009) to ensure stability when integrating the modified version of the ODE in Equation 2. Our method saves computational cost by permitting a comparatively large default time-step ( $\Delta t_{\rm cond} \sim 1\,{\rm s}$ ) and very often keeping the number of Newton-Raphson iterations below three. Explicitly, given the time-step for condensation/evaporation,  $\Delta t_{\rm cond}$ , and the notation  $R_i(t^n) = R_i^n$ , we make the approximation  $f_{\rm v}(R) \approx f_{\rm v}(R_i^n)$  and let  $f_{\rm v}(R_i^n) = f_{\rm v}$ , the ODE is discretised as

$$0 = \frac{Z - (R_i^n)^2}{\Delta t_{\text{cond}}} - \frac{2f_{\text{v}}[(S-1) - aZ^{-1/2} + bZ^{-3/2}]}{F_{\text{k}} + F_{\text{d}}};$$
(4)

where  $Z = (R_i^{n+1})^2$ . The Newton-Raphson method is then used to find the real positive root of the polynomial,

$$g(Z) = \frac{Z - (R_i^n)^2}{\Delta t_{\text{cond}}} - \frac{2f_{\text{v}}[(S-1) - aZ^{-1/2} + bZ^{-3/2}]}{F_{\text{k}} + F_{\text{d}}};$$
(5)

such that the radius of each superdroplet at the subsequent time-step is given by  $R_i^{n+1} = +\sqrt{Z}$  when g(Z) = 0. We use g(Z) rather than the higher order polynomial in terms of  $+\sqrt{Z}$  because it converges more rapidly — in fact in the limit a = b = 0, the Newton-Raphson method can find the true root of Equation 5 with one iteration. A reasonable initial guess close to the true solution for g(Z) = 0 can make convergence fast and so we choose our first guess for Z to be  $(R_i^n)^2$  unless  $S > 1 + \sqrt{\frac{4a^3}{27b}}$ , in which case the superdroplet is already activated and we make the initial guess very large: at least  $10^{-6} \, \mathrm{m}^2$ .

There are upto three real roots of g(Z) and so we must ensure that the genuine solution for  $R_i^{n+1}$  is converged upon. If the droplet was previously and is currently un-activated, and in an environment with a supersaturation less than its activation supersaturation, i.e. if

$$S \le 1 + \sqrt{\frac{4a^3}{27b}}, \quad \text{and} \quad R_i^n < \sqrt{\frac{3b}{a}}, \tag{6}$$

then the uniqueness of the solution to q(Z) = 0 is guaranteed. Likewise, if the time-step is small enough,

115 
$$\Delta t_{\text{cond}} \le \frac{25b(F_{\text{k}} + F_{\text{d}})}{2a^2 f_{\text{v}}} \sqrt{\frac{5b}{a}},$$
 (7)

uniqueness is guaranteed (as in Matsushima et al. (2023) modulo the ventilation factor). In either of these cases we attempt two iterations of the Newton-Raphson method and then perform a standard local error test for convergence, whereby the method has converged if  $g(Z) < \alpha + \beta g\left(R_i^n\right)^2$  for some absolute and relative tolerances,  $\alpha$  and  $\beta$ , respectively  $R_i^{n+1}$  has been found if the test passes, whereas if it fails we perform further iterations and check for convergence after each one. However, to prevent parasitic cases from running for infinite time, the simulation is terminated if a chosen maximum number of iterations is exceeded R. When neither condition for a unique solution is met adaptive time-stepping is performed, whereby R divided into sub-steps which each obey the second criteria for uniqueness. It is highly improbable but these sub-steps can be extremely small (Matsushima et al., 2023). We therefore allow a minimum sub-step to be set in order to reduce the cost of the simulation but at the risk of finding the incorrect solution for  $R_i^{n+1}$ .

## 125 3 Collisions

120

130

Modelling collisions between droplets in CLEO can be done with or without collisional breakup and rebound. Collisional breakup during rainfall has been observed in situ and has been suggested as an important control on the timing and intensity of rain in LES (Seifert et al., 2005; Testik and Rahman, 2017). Since breakup converts large droplets into significantly smaller ones, its effect on the higher moments of the droplet size distribution can be appreciable (e.g. McFarquhar, 2004) (references in Morrison et al., 2020). This motivates two alternative treatments for collisions in CLEO, schematised in Figure 2, which are:

- (a) the original algorithm of Shima et al. (2009) which models only collision-coalescence,
- (b) the extended algorithm which also includes rebound and breakup.

 $<sup>^{1}</sup>$ By default we choose  $\alpha=0.01$  and  $\beta=0.0$ , and the maximum number of iterations as 50.

**Figure 2.** a) Schematic of the original collision-coalescence algorithm from Shima et al. (2009). b) The extended collision algorithm including breakup and rebound as well as coalescence. A lightening/darkening of a superdroplet indicates a decrease/increase in its multiplicity.

In both treatments, the steps to determine whether or not a collision occurs is the same as in Shima et al. (2009). All the superdroplets in a grid-box are first randomly paired with one-another. Then for each pair the probability that they collide is compared to a random number,  $\phi_{\alpha}$ , to determine whether or not a collision is enacted. The probability that two superdroplets collide,  $p_{\alpha}$ , is the probability that two real droplets collide,  $P_{jk}$ , scaled by a factor dependent on the superdroplets' multiplicities. Usually  $P_{jk}$  can be calculated given a collision kernel,  $K_{jk}$ , as

$$P_{jk} = K_{jk} \frac{\Delta t_{\text{coll}}}{\Delta V},\tag{8}$$

for a pair of droplets j and k, where  $\Delta t_{\rm coll}$  and  $\Delta V$  are the collision time interval and volume, respectively (Shima et al., 2009). The probability two superdroplets collide is then

$$p_{\alpha} = \max(\xi_j, \xi_k) \frac{n_{\rm s}(n_{\rm s} - 1)/2}{\lfloor n_{\rm s}/2 \rfloor} P_{jk},$$
 (9)

where  $n_{\rm s}$  is the number of superdroplets in the collision volume,  $\Delta V$ .

CLEO defines the collision probability flexibly to make it easy to switch between different formulations of  $P_{jk}$ . There are large uncertainties in determining the collision probability of a pair of superdroplets' because the collision kernel is so poorly constrained. As such, several formulations for  $K_{jk}$  exist, which differ for example in their treatment of turbulence (e.g. Long, 1974; Hall, 1980; Ayala et al., 2008). The calculation for  $P_{jk}$  in CLEO can therefore be changed before compilation as long as it obeys the C++20 concept, (ISO, 2020) (see also Bayley et al., 2025a), we prescribe for the collision probability. There are two  $P_{jk}$  calculations implemented at the time of writing: one for Golovin's kernel (Golovin, 1963), and the other for the hydrodynamic kernel with the formulation for the terminal velocity from Simmel et al. (2002) and collision efficiencies from Long (1974), as also done by Grabowski and Wang (2009).




If a collision occurs, treatments (a) and (b) differ in their outcome. In the original algorithm, coalescence is assumed and the superdroplets are updated as in Section 3.1. Whereas if a collision occurs in the extended algorithm, the last step of the original collision-coalescence algorithm is modified as shown in Figure 2b. Rather than assuming a successful collision of two superdroplets results in coalescence, the attributes of the original superdroplets are used to determine if the collision causes rebound, coalescence, or breakup. We then modify the superdroplets to reflect the appropriate outcome: as in Section 3.1 for coalescence and as in Section 3.2 for breakup. In the event of rebound, the superdroplets remain unchanged.

CLEO has several options for how the attributes of colliding superdroplets determine whether the outcome of the extended algorithm is rebound, coalescence, or breakup. This is physically motivated since properties of droplets, such as their velocities, shapes, and masses, determine whether a collision has sufficient energy to result in coalescence or breakup instead of rebound, but exactly how such properties determine the outcome of a collision is disputed. Our algorithm can therefore easily interchange various formulations from the literature, at the time of writing from Low and List (1982a), Testik et al. (2011), and Szakáll and Urbich (2018).

As an example, Figure 3 demonstrates how the outcome of a collision is determined based on Testik et al. (2011). There are three possible regimes depending on the magnitude of the collision kinetic energy,  $T_{\rm E}$ , relative to the surface tension energy of the smaller and larger droplet,  $S_S$  and  $S_L$ , respectively (Testik, 2009). Using the definitions

$$T_{\rm E} = \frac{2}{3} \pi \rho_l \frac{R_j^3 R_k^3}{R_i^3 + R_k^3} |\mathbf{v}_{j,\infty} - \mathbf{v}_{k,\infty}|^2, \tag{10}$$

$$S_i = 4\pi\sigma_l R_i^2,\tag{11}$$

$$R_S = \min(R_i, R_k),\tag{12}$$

$$R_L = \max(R_i, R_k),\tag{13}$$

where  $\rho_l$  is the density and  $\sigma_l$  is the surface tension of liquid water, these three regimes are:

- 1. when  $T_{\rm E} < S_S$  either coalescence or rebound occurs,
- 2. when  $S_S < T_{\rm E} < S_L$  either coalescence or breakup occurs,
- 3. when  $T_{\rm E} > S_L$  breakup occurs.

In regimes 1 and 2 a coalescence efficiency,  $E_c \in [0,1]$ , gives the probability two droplets coalesce,  $P_{c,jk}$ , given that they collided, i.e.  $P_{c,jk} = E_c P_{jk}$ . Here, the parametrisation for  $E_c$  is

$$E_{\rm c} = e^{-1.15W},$$
 (14)

where the Weber number,  $W = \frac{T_{\rm E}}{S_{\rm c}}$ , and  $S_{\rm c} = 4\pi\sigma(R_S^3 + R_L^3)^{2/3}$ ; as argued for by Straub et al. (2010). Unlike de Jong et al. (2022), we rescale the original random number which was used to determine that a collision occurred rather than draw a new random number,  $\phi_{\alpha}' = \phi_{\alpha}/(\lfloor p_{\alpha} \rfloor - p_{\alpha})$ , and then we compare  $\phi_{\alpha}'$  with  $E_{\rm c}$  to decide if the collision results in coalescence or the alternative.

**Figure 3.** The probability, based on Testik et al. (2011), for whether a collision between two droplets results in coalescence, rebound or breakup. Inside the blue dashed contour it is certain that a collision results in breakup, between the blue and purple contours it may result in coalescence or breakup, and outside the purple contour it may result in coalescence or rebound. The colour gives the coalescence efficiency, in other words the probability of coalescence given that a collision occurs, according to Straub et al. (2010).

Figure 4. Coalescence as in Shima et al. (2009). a) The superdroplet representation of coalescence whereby the multiplicity of the more multiplicitous droplet decreases to increase the size of the other superdroplet. A lightening/darkening of a superdroplet indicates a decrease/increase in its multiplicity. b) The real droplet equivalent of the superdroplets above when  $\xi_j = 3$ ,  $\xi_k = 2$  and  $\tilde{\gamma}_{\alpha} = 1$ .

# 3.1 Coalescence

The change in attributes of a pair of superdroplets that undergo coalescence follows Shima et al. (2009) and is illustrated in Figure 4.

Given a pair of superdroplets (j,k) we can choose without loss of generality the multiplicity of the  $j^{\text{th}}$  superdroplet to be at least as large as that of the  $k^{\text{th}}$ , i.e.  $\xi_j \geq \xi_k$ . The permitted number of consecutive coalescence events,  $\tilde{\gamma}_{\alpha}$ , is then used to

determine how the superdroplet attributes are changed to enact coalescence. To calculate  $\tilde{\gamma}_{\alpha}$ , we compare  $p_{\alpha}$  to the random number  $\phi_{\alpha}$  as well as the multiplicities of the superdroplets since

$$\gamma_{\alpha} = \begin{cases}
\lfloor p_{\alpha} \rfloor + 1 & \text{if } \phi_{\alpha} < p_{\alpha} - \lfloor p_{\alpha} \rfloor, \\
\lfloor p_{\alpha} \rfloor & \text{if } \phi_{\alpha} \ge p_{\alpha} - \lfloor p_{\alpha} \rfloor;
\end{cases}$$
(15)

$$\tilde{\gamma}_{\alpha} = \min(\gamma_{\alpha}, \lfloor \xi_j / \xi_k \rfloor).$$
 (16)

There are then two possible scenarios which enact coalescence:

(a) if  $\xi_j > \tilde{\gamma}_{\alpha} \xi_k$ , the less multiplications superdroplet grows by consuming  $\tilde{\gamma}_{\alpha} \xi_k$  droplets from the other superdroplet,

$$\xi_i' = \xi_i - \tilde{\gamma}_\alpha \xi_k, \ \xi_k' = \xi_k, \tag{17}$$

$$R'_{j} = R_{j}, R'_{k} = (\tilde{\gamma}_{\alpha} R_{j}^{3} + R_{k}^{3})^{1/3},$$
 (18)

$$M'_{s,i} = M_{s,i}, M'_{s,k} = (\tilde{\gamma}_{\alpha} M_{s,i} + M_{s,k}),$$
 (19)

$$a_{j}^{\prime\prime} = a_{j}^{\prime}, \ a_{k}^{\prime\prime} = a_{k}^{\prime};$$
 (20)

(b) otherwise  $\xi_j = \tilde{\gamma}_\alpha \xi_k$ , and rather than ending up with  $\xi_j' = 0$  both superdroplets are used to represent coalesced droplets,

$$\xi_i' = \lfloor \xi_k/2 \rfloor, \ \xi_k' = \xi_k - \xi_i', \tag{21}$$

$$R'_{j} = R'_{k} = (\tilde{\gamma}_{\alpha} R_{j}^{3} + R_{k}^{3})^{1/3}, \tag{22}$$

$$M'_{s,j} = M'_{s,k} = (\tilde{\gamma}_{\alpha} M_{s,j} + M_{s,k}),$$
 (23)

$$a_{j}^{\prime\prime} = a_{j}^{\prime}, \ a_{k}^{\prime\prime} = a_{k}^{\prime};$$
 (24)

In the exceptional case that  $\xi_j'=0$ , we currently choose to terminate the simulation rather than remove the superdroplet.

# 3.2 Breakup





To keep the computational advantages of the original SDM collision algorithm, the fragments created from one breakup event are represented by a single superdroplet. In reality, collisional breakup results in a spectrum of different fragments whose likelihood depends on the two droplets that collided. However the number of simulated particles must be unchanged during collisions to ensure SDM is manageable, and thus the fragments from breakup must all have the same attributes. Ideally these would be randomly sampled from a known fragment distribution, as in de Jong et al. (2022) and Bringi et al. (2020). However observations of these distributions are limited by both resolution and the undersampling of initial droplet sizes (Low and List, 1982a, b; Schlottke et al., 2010; Prat et al., 2012; Szakáll and Urbich, 2018).

Given the acute shortage of empirical data and our desire for high computational performance, our algorithm does not randomly sample a known fragment distribution. Instead the properties describing the fragments from collisional breakup,  $\xi_{\rm frag}$ ,  $R_{\rm frag}$  and  $M_{s,{\rm frag}}$ , are determined by the initial superdroplet attributes. We provide several ways for how to do this in CLEO; for example, one way asserts mass conservation and that the number of fragments depends on a collision's kinetic energy based on Schlottke et al. (2010), as detailed in Appendix A. Such a method does not seek to reproduce the measured fragment distributions, but rather pursue an overtly simple approach with the intention of exploring whether and how breakup could impact observables such as rain evaporation rates and radar reflectivity signals (e.g. Morrison et al., 2020; Bayley et al., 2025b).

Two previous papers have introduced a collisional breakup algorithm into SDM (Bringi et al., 2020; de Jong et al., 2022). Unlike the breakup algorithm in McSnow (Bringi et al., 2020), we ensure the conservation of superdroplet number and thus computational tractability. Our algorithm is similar to that of pySDM (de Jong et al., 2022), but with two adjustments. Firstly, whether a collision results in rebound, coalescence, or breakup is deterministic in our algorithm, whereas an additional probabilistic step is used in de Jong et al. (2022). Secondly, in the event of breakup we avoid a recursive algorithm by prohibiting multiple breakup events in a single time-step. In the original algorithm of Shima et al. (2009), consecutive coalescence events can be enacted in a single time-step via the factor for the permitted number of consecutive coalescence events,  $\tilde{\gamma}_{\alpha}$ . Allowing consecutive coalescence rests on the assumption that after the first coalescence event(s) the droplets are similar enough that the coalescence probability is approximately unchanged, meaning the collision probability and its outcome do not need recalculation before subsequent events. In our breakup algorithm we make the opposite assumption: that after a single breakup event there is negligible probability of the resultant droplets undergoing further collisional breakup with one-another and thus only one breakup event can occur per time-step ( $\tilde{\gamma}_{\alpha}=1$ ). This is a reasonable assumption for the O(1s) time-step required for collision-coalescence and is a necessary reduction to the computational cost of the algorithm in large simulations.

The change in superdroplets that undergo collisional breakup is illustrated in Figure 5 and has two cases analogous to coalescence:

(a) if  $\xi_j \neq \xi_k$ , we can choose without loss of generality  $\xi_j > \xi_k$  such that  $\xi_k$  droplets from the more multiplications superdroplet are involved in the breakup with the other superdroplet,

$$\xi_j' = \xi_j - \xi_k, \ \xi_k' = \xi_{\text{frag}},\tag{25}$$

Figure 5. a) The superdroplet representation of breakup decreases the multiplicity of the more multiplicitous droplet to fragment the other droplet. A lightening/darkening of a superdroplet indicates a decrease/increase in its multiplicity. b) The real droplet equivalent of the superdroplets above when  $\xi_j = 3$ ,  $\xi_k = 2$ , and  $\xi_{\text{frag}} = 10$ .

$$R_j' = R_j, \ R_k' = R_{\text{frag}},\tag{26}$$

$$M'_{s,j} = M_{s,j}, M'_{s,k} = M_{s,frag},$$
 (27)

$$a_{j}^{\prime\prime} = a_{j}^{\prime}, \ a_{k}^{\prime\prime} = a_{k}^{\prime};$$
 (28)

(b) if  $\xi_j = \xi_k$ , both superdroplets are used to represent the result of breakup,

$$\xi_j' = \operatorname{round}(\xi_{\text{frag}}/2), \ \xi_k' = \xi_k - \operatorname{round}(\xi_{\text{frag}}/2), \tag{29}$$

$$R_i' = R_k' = R_{\text{frag}},\tag{30}$$

$$M'_{s,j} = M'_{s,k} = M_{s,\text{frag}},\tag{31}$$

$$a_{j}^{\prime\prime} = a_{j}^{\prime}, \ a_{k}^{\prime\prime} = a_{k}^{\prime};$$
 (32)

#### 4 Motion



We use a simple predictor-corrector method to model superdroplet motion. One of the most computationally expensive parts to Lagrangian microphysics is the motion of particles throughout the domain (Matsushima et al., 2023; Bayley et al., 2025a). However, since a superdroplet's location is not the true location of the droplets it represents, there is no imperative to precisely resolve a superdroplet's position. Indeed looking at the conceptual picture of SDM, perhaps the most suitable choice for superdroplet motion would be probabilistic, similar to Curtis et al. (2017). However, we save expense by choosing Heun's method, a second-order predictor-corrector method, with a simple linear interpolation of the wind velocity. This method by construction preserves the divergence of the flow field, and gives physically consistent results at a 1s time-step (Grabowski et al., 2018).

In summary, we integrate the equation for the coordinates  $x_i$  of the  $i^{\mathrm{th}}$  superdroplet,

$$\frac{\mathrm{d}\boldsymbol{x_i}(t)}{\mathrm{d}t} = \boldsymbol{u_i}(\boldsymbol{x_i}, t),\tag{33}$$

where the superdroplet's velocity,  $u_i(x_i,t)$ , is

$$u_i(x_i,t) = w(x_i,t) + v_{i,\infty}(x_i), \tag{34}$$

and  $v_{i,\infty}(x_i) = -v_{i,\infty}(x_i)\hat{z}$  is the superdroplet's terminal velocity, and  $w(x_i,t)$  is the wind velocity, obtained by simple linear interpolation of the wind velocity at a grid-box's faces. In CLEO the terminal velocity is defined by a C++20 concept so that different formulations can be easily interchanged (ISO, 2020) (see also Bayley et al., 2025a). The options currently available are depicted in Figure 6. Using the notation  $x_i(t^n) = x_i^n$ , and the time-step for superdroplet motion,  $\Delta t_{\rm m}$ , the predicted coordinates at the subsequent time-step are

$$\tilde{\boldsymbol{x}}_{i}^{n+1} = \boldsymbol{x}_{i}^{n} + \boldsymbol{u}_{i}^{n}(\boldsymbol{x}_{i}^{n}) \Delta t_{m}, \tag{35}$$

which are corrected to

$$\boldsymbol{x_i^{n+1}} = \boldsymbol{x_i^n} + \left[\boldsymbol{u_i^n(\tilde{\boldsymbol{x}}_i^{n+1})} + \boldsymbol{u_i^n(\boldsymbol{x_i^n})}\right] \frac{\Delta t_{\text{m}}}{2}.$$
(36)

This formulation for particle motion could easily be extended to account for the effects of particle inertia and sub-grid scale turbulence, for example following Naumann and Seifert (2015), which would increase the fidelity of the particle trajectories, especially for large droplets whose inertial relaxation timescale is larger than Kolmogorov time scale.

# 5 Validations


This section presents three test-cases which demonstrate the behvaiour of our numerical methods for condensation/evaporation, collisions between droplets, and droplet motion individually. Furthermore, we present a final integrated test-case applying all three methods at once using the 1-D Kinematic Driver (KiD) modelling framework (Shipway and Hill, 2011, 2012).




Figure 6. The formulations for terminal velocity currently available in CLEO, and for comparison the formulation according to Pruppacher and Klett (1978) at  $T = (288.15 \pm 15.00)$  K in a standard atmosphere.

Figure 7 shows the results from CLEO for the same test-case as described in Section 7 of Arabas and Shima (2017) for the adiabatic expansion/contraction of a rising/falling volume of air through a hydrostatically equilibrated atmosphere. We used 0.0 and 0.01 for the relative and absolute tolerances of our ODE solver, and 50 and 1 ms for the maximum number of Newton-Raphson iterations and minimum sub-time-step, respectively. Each column shows the evolution of the air mass given a certain number concentration and/or initial radius of its dry aerosol mono-size distribution. In particular, it shows how the displacement, supersaturation, and droplet radii evolve given three different mean vertical velocities,  $\langle w \rangle$ , for the ascent/descent of the air mass. The first two columns show that, as expected, the faster the vertical velocity, the higher the dis-equilibrium supersaturation during ascent (and the lower the dis-equilibrium supersaturation during descent). By comparing the first two columns, we can also see that increasing the aerosol concentration lowers the dis-equilibrium supersaturation. All these trends in the supersaturation can be explained by faster vertical velocities and/or fewer aerosol causing the rate at which the relative humidity increases with height to be greater than the rate at which the droplets undergo condensation. Conversely, the lower the vertical velocity and higher the droplet number concentration, the closer the air mass to its equilibrium behaviour, where supersaturation does not exceed 0% and the radii follow the Köhler curve. In the third column however, the Köhler curve is not approached, because the radii and number concentration are low enough that cusp bifurcation occurs and a sudden "jump" in the droplet growth is observed instead. Due to differing numerics and initial thermodynamic conditions compared to Arabas and Shima (2017), the plots are not completely identical (we use 1000 hPa, 298.15 K, and 98.0%, for pressure, temperature and relative humidity, respectively), but our supersaturations are within 0.1% of theirs and we also observe cusp bifurcation, albeit at a slightly smaller radius of 0.03 µm. Notwithstanding these differences, we can say that CLEO is able to reproduce the expected evolution of microphysical and thermodynamic conditions for all three mean vertical velocities.

Figure 8 shows the results from CLEO for the same test-case as described in Section 5.1.4 of Shima et al. (2009) for the stochastic collisions between water droplets in an arbitrary volume of air. We show results for the same setups as Shima et al. (2009), namely for the evolution of the mass density distribution (normalised by log-space bin-width) given different






collision kernels and droplet populations whose volumes are initially exponentially distributed. Figure 8a shows the evolution when the kernel from Golovin (1963) is used, whereas Figures 8b-c show the evolution when the hydrodynamic kernel is used. For all cases, we reproduce the results of Shima et al. (2009) when collisions are assumed to result in coalescence. We also demonstrate in Figures 8b-c how the distribution evolves according to the extended framework from Section 3, which also allows for collisional breakup and rebound. The probability of breakup and rebound was calculated from Testik et al. (2011) and Straub et al. (2010) as described in Section 3, and if breakup occurred the fragment properties was determined based on the parametrisation of Schlottke et al. (2010) (as described in Appendix A). Up-to about 600s the evolution of the droplet size distribution is comparable to the coalescence-only case, but the subsequent growth of droplets is much delayed and converges to a stationary distribution centred at 750 µm, rather than 2400 µm in the coalescence-only case. As expected, collisional breakup slows and inhibits the growth of mm-size droplets. It also has a much larger impact on the distribution's evolution than changing the collision efficiencies from Long (1974) to Hall (1980) in the coalescence-only case. It remains to be seen how a kernel accounting for turbulent effects would behave, but that too would likely show considerably larger differences (Grabowski and Wang, 2009).

Figure 9 shows the results from CLEO for the motion of particles in the 2-D kinematic laminar flow model described in Section 2.1 of Arabas et al. (2015). 128 superdroplets are initially randomly distributed throughout each grid-box in the domain and the flow is simulated for 1 hour with a time-step of 1s for both motion and data output, and for three simulations with grid-spacing  $\Delta x = \Delta z = 100 \,\mathrm{m}$ ,  $50 \,\mathrm{m}$ , and  $25 \,\mathrm{m}$ , respectively. We do not simulate microphysics and the droplets have no fall velocity, so the superdroplets behave like tracers for the flow. As expected, CLEO's particle motion does indeed follow the flow field and respect its divergence-free properties, as confirmed by the conservation of particle number in each grid-box throughout the simulations (not shown). Figure 9 also verifies that increasing the grid resolution increases the accuracy of the particle motion because the interpolation distances in the predictor-corrector algorithm reduce.

Figure 10 demonstrates the integrated test of CLEO's numerical methods using the 1-D KiD framework. Here we setup the column as described by Shipway and Hill (2012), with a 25 m grid-spacing and sinusoidal updraught velocity for the first 10 min with the maximum speed constant,  $w_1 = 3 \,\mathrm{m/s}$ . We couple CLEO via python bindings to PyMPDATA (Bartman et al., 2022) to advect water vapour, whereas we advect liquid water (superdroplets) within CLEO given the wind field at each time-step (1.25 s). We show results for a 10 member ensemble where for each member we initialise  $50 \,\mathrm{cm}^{-3}$  of dry aerosol throughout the domain by assigning 256 superdroplets per grid-box each a constant multiplicity and randomly sampling a single-mode lognormal distribution with a geometric mean diameter of 0.08 mm and a log standard deviation of 1.4, as chosen by Hill et al. (2023). Unlike both Shipway and Hill (2012) and Hill et al. (2023), we specify the initial air density profile to be consistent with hydrostatic equilibrium including the initial water vapour, with the surface pressure at  $1000 \,\mathrm{hPa}$ . The different microphysics schemes and density profiles causes slight differences between our clouds and those of Shipway and Hill (2012), Hill et al. (2023) and de Jong et al. (2022), but nevertheless the qualitative behaviour of the test-cases are in agreement and our results lie within the spread amongst different SDMs in Hill et al. (2023). Figure 10(a-b) shows the liquid water mass mixing ratio,  $q_t$ , when only condensation/evaporation is modelled. The cloud reaches a steady state immediately after the updraught velocity terminates, with liquid water content peaking in the middle of our cloud, albeit at a higher value and







lower height. Without collision-coalescence, there is no spread across the ensemble members, which indicates little sensitivity of condensation/evaporation to the sampling of the initial dry aerosol distribution, as reported by Hill et al. (2023). In contrast, Figure 10(c-d) show the same setup when not only condensation/evaporation, but also collision-coalescence and droplet motion (including their fall velocities) are modelled, and in these simulations the ensemble spread is significant because of the stochasticity in the collision-coalescence algorithm. The mean precipitation onset is around 30 min, in agreement with the range of SDMs in Hill et al. (2023), and likewise the evolution of the liquid water path, cloud droplet number concentration, mean volume diameter and its standard deviation also lie within the SDMs' spread (not shown). More exact agreement is hard to achieve given the ambiguity in the test-case setup and the large model spread reported by Hill et al. (2023), for which further inter-model comparison would be needed to more exactly attribute the discrepancies. Nevertheless, CLEO does indeed behave as expected for a SDM and is in agreement with other microphysics models for warm rain within the KiD framework.

### 6 Conclusions

CLEO is a concise and fully-functioning SDM for warm-cloud microphysics. We have validated our SDM against known test-cases for condensation/evaporation, collisions between droplets, and Lagrangian particle motion, including an integrated test including all three using the 1-D KiD framework. The motion is less expensive than higher-order methods yet still preserves the flow-field, is non-diffusive, and has increasing accuracy with increasing grid resolution. For condensation/evaporation we solve the Köhler theory ODE explicitly. Our method incorporates ventilation effects and droplet (re)activation as well as a number of cost-saving measures including adaptive sub-time-stepping. For collisions between droplets, CLEO has various options which reflect the uncertainty in the current knowledge about collision probabilities and their outcomes. CLEO can easily switch between different formulations of the collision kernel and, motivated by a need to investigate the possible influence of collisional breakup on rain evaporation and downdraughts, CLEO includes a framework to model collisional breakup and rebound. We do this in a way that still respects the key features of the original collision-coalescence algorithm by conserving superdroplet number and decreasing the variance in the collisions with decreasing superdroplet multiplicity.

CLEO is useful to a broad group of researchers that require a model for warm-cloud microphysics. Simulations can be up-to 3-D and can include any combination of the primary microphysical processes behind warm-rain, each with their own independent time-step. CLEO can be stand-alone, perform "piggybacking" of LES (Grabowski, 2014), or be used as a fully fledged microphysics scheme. Like other SDMs, CLEO provides a precise representation of the droplet size distribution and an intuitive depiction of microphysical processes. We have also made sensitivity studies easy to conduct through CLEO's high-degree of flexibility, whereby microphysics and data output can be easily customised in order to assess the impact of individual microphysical processes (Bayley et al., 2025a), potentially for probing microphysical uncertainties. Finally, since CLEO is also well-suited to high-performance computers, it has the potential to be used in more demanding simulations which probe how cloud microphysics and its uncertainties propagate to larger spatial and temporal scales.

*Code availability.* CLEO is published on it's GitHub page: https://github.com/yoctoyotta1024/CLEO, alongside it's documentation: https://yoctoyotta1024.github.io/CLEO. Version v0.52.0 is described and tested in this paper.

Code and data availability. All the code and results included this paper can be accessed from the dataset: Bayley (2025).

# Appendix A: Example Parametrisation for Fragments From Collision Breakup

Here we expound one of CLEO's options for how to determine the properties of a superdroplet which represents the fragments from collisional breakup; namely how to define  $\xi_{\text{frag}}$ ,  $R_{\text{frag}}$  and  $M_{s,\text{frag}}$  in Section 3.2 for Equations 25 to 32. As a simple approach for exploring the potential importance of collisional breakup in warm-cloud microphysics, here we assert mass conservation and that the total number of fragments,  $N_{\text{frag}}$ , depends on a collision's kinetic energy,  $T_{\text{E}}$ . Schlottke et al. (2010) reported that  $N_{\text{frag}}$  depends on  $T_{\text{E}}$  as:

$$N_{\text{frag}} = \left(\frac{3}{2} - (T_{\text{E}}/\mu \text{J})^{0.135}\right)^{-1}.$$
 (A1)

Since this equation diverges at  $T_{\rm E}/\mu \rm J = 1.5^{\frac{1}{0.135}}$ , we limit  $N_{\rm frag} \le 25$  for  $T_{\rm E}/\mu \rm J > 16.50$ ; and so that collisional breakup always results in more droplets than collided, even when  $\xi_j = \xi_k = 1$ , we impose the condition that  $N_{\rm frag} \ge 2.5$ . Both of these amendments are concordant with the experimental data presented by Schlottke et al. (2010).

Given  $N_{\text{frag}}$  is the number of fragments produced by a single (real) droplet breakup event, the superdroplet multiplicity must be as follows,

$$\xi_{\text{frag}} = \text{round}(\xi_k * N_{\text{frag}}),$$
 (A2)

and therefore, to ensure mass conservation,

$$R_{\text{frag}} = \left[ \frac{\xi_k}{\xi_{\text{frag}}} (R_j^3 + R_k^3) \right]^{1/3},$$
 (A3)

and


380

385

$$M_{s,\text{frag}} = \left[\frac{\xi_k}{\xi_{\text{frag}}} (M_{s,j} + M_{s,k})\right]. \tag{A4}$$

Author contributions. CJAB is the creator of CLEO and main developer, she also wrote and edited the manuscript. AKN and RV supervised the project, providing direction and support, as well as teaching CJAB about various aspects of cloud physics included in this paper. BS conceptualised the project, and had many discussions with CJAB which shaped the writing of the paper and provided scientific and technical support. He also helped analyse the test case results. FP contributed the parametrisation for the ventilation factor in condensation/evaporation. SIS supervised CLEO's development, and contributed ideas and advice to the numerical methods for droplet breakup and condensation/evaporation. SIS also helped design the test cases for the numerical methods and analyse the results. AKN, FP, RV, BS and SIS all gave extensive feedback which contributed to the writing of the manuscript.

405

410

Competing interests. The contact author has declared that none of the authors has any competing interests.

Acknowledgements. C. J.A. Bayley thanks Sylwester Arabas (AGH University of Krakow, Poland) and Emma Ware (University of California, Davis, USA) for their engaging discussions and technical support setting up the 1-D KiD model using PyMPDATA. We gratefully acknowledge code contributions to CLEO from Sergey Kosukhin and Lukas Kluft (Max Planck Institute for Meteorology, Germany; MPI-M). A special thanks is given to Yvonne Schrader (MPI-M) for her excellent advice on the graphic designs in this paper, as well as Romain Fiévet (MPI-M) for conducting the MPI-M internal review.

A. K. Naumann, R. Vogel and F. Poydenot have received funding which supported this work from the Deutsche Forschungsgemeinschaft (DFG, German Research Foundation) under Germany's Excellence Strategy - EXC 2037 "Climate, Climatic Change, and Society" (project number 390683824). R. Vogel further acknowledges support from an ERC starting grant (ROTOR, grant no. 101116282). This project has received funding from Horizon Europe programme under Grant Agreement No 101137680 via project CERTAINTY (Cloud-aERosol inTeractions & their impActs IN The earth sYstem). The authors further express their appreciation for the work of the developers of the free and open-source software which underlies CLEO, especially from the developers of Git, GitHub, Python, the C++ standard libraries, and, above all, Kokkos. We also thank the Gesellschaft für wissenschaftliche Datenverarbetitung mbH Göttingen (GWDG) from the information and communication services CLEO's development has benefited from, and finally we thank the Deutsche Klimarechenzentrum (DKRZ) for the computer facilities from project 1183 we used to conduct this work.

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

Figure 7. Test of condensation/evaporation as in Arabas and Shima (2017, Sect.7, Figure 5). The panels are the same as in Arabas and Shima (2017), except the radius in the third column is 0.02 μm smaller, because slightly different numerics and initial thermodynamic conditions affect the radius at which cusp bifurcation is observed. Each column shows the evolution of an air parcel as it adiabatically rises and falls with a certain number concentration and initial radius of mono-distributed dry aerosol. Top row: the displacement of the parcel against its supersaturation; Middle row: The supersaturation as a function of the droplet radii; Bottom row: the displacement of the parcel against the droplet radii.

Figure 8. Test of SDM collisions as in Shima et al. (2009, Sec. 5.1.4, Figure 2). The line-style indicates the collision kernel; the line-width indicates the number of superdroplets, N; the colour indicates the time. Sub-figure a) additionally shows the analytical solution for the Golovin kernel at each time (solid grey). The sub-figures are the same as in Shima et al. (2009) except in panels b) and c), where, as well as the hydrodynamic kernel, we additionally plot the distribution's evolution when breakup and rebound are accounted for, based on the parametrisations from Schlottke et al. (2010), Straub et al. (2010), and Testik et al. (2011) (see main text, Section 3).

**Figure 9.** Results of tracer particle motion in the 2-D divergence free laminar flow described by Arabas et al. (2015). Grid-spacing decreases from top to bottom panel.

**Figure 10.** Evolution of the liquid water mass mixing ratio,  $q_t$ , during the 1-D KiD test-case: (a-b) only modelling condensation/evaporation, (c-d) modelling condensation/evaporation, collision-coalescence and droplet sedimentation. Results are shown for the mean and interquartile range over an ensemble of 10 members, and figures b) and d) are cross sections across the vertical coloured lines in a) and b) respectively.