# Peer review of "CLEO: The Numerical Methods of a New Superdroplet Model including a Droplet Breakup Algorithm"

_EGUsphere, 2025_

## Referee Comment (RC1)

Bayley et al. provide a valuable introduction and description of the detailed parameterization choices and numerical methods for warm rain processes in their new SDM, "CLEO." The included thorough description of numerics for condensation and other processes makes this manuscript valuable for future replication and comparison of CLEO-based results with other SDM methods in future work. I have some small objections to claims and language used to describe Eulerian microphysics in the introduction, as well as some suggestions to validate and/or demonstrate the more novel aspects of CLEO (ventilation, breakup) that would strengthen the impact of this paper. Once addressed, I believe this manuscript will be a valuable addition to the scientific model description literature.

**Major Comments:**

1.  The abstract and introduction include strong and potentially misleading descriptions of traditional microphysics methods that would benefit from a more nuanced presentation. Specifically:
    - The abstract mentions "Eulerian models" but should be careful to distinguish microphysics parameterizations from the entire class of "Eulerian" models; all ESMs are Eulerian in their treatment of momentum, after all.
    - Bold claims like "their underlying assumptions are inconsistent with even the most basic kinematic processes" need elaboration or tempering
    - Claims that SDM converges toward the Smoluchowski equation while bulk or bin schemes do not are needs elaboration. Consider that linear sampling in the SDM would lead to slower than expected convergence, wherease bin schemes provide a direct finite element approximation to the Smoluchowski equation.
    - The introduction should recognize that the rationale for developing bulk and bin microphysics largely stems from computational limitations that necessitate approximations; the SDM is still infeasible for global simulation, thus we will continue to rely on bulk microphysics. In fact, many non-Lagrangian approaches are based on clever mathematical approximations and have enabled broad advances in atmospheric science based on necessary complexity tradeoffs. Lagrangian microphysics is itself not free of assumptions or limitations, nor differences between implementations (see Hill 2023) and these should be reasonably discussed in similar level of detail.
2.  Some of the new additions to CLEO (ventilation, breakup) have not been rigorously validated in the included experiments.

- o Ventilation was not validated for expected qualitative behavior (cooling, evaporation of rain) in any of the example cases. Please show an instance where the ventilation effect can be qualitatively observed.
- o The choice to eschew fragment size distributions altogether is a break from the norm, and warrants further discussion and validation. I recommend including a comparison of the stationary PSD that results from this choice of fragment size with those of traditional implementations (Straub 2010, McFarquhar) to reveal any systematic biases that that your implementation may suffer, such as overproduction of cloud-size fragments.
- o Figure 10: Including a plot of CLEO's predictions of precipitation rate and timing would further be extremely useful for comparison with other SDMs in the Hill 2023 study (and would back up the claim in L350-351).
- o Seeing a verification example where the terminal velocity is included in addition to the passive tracer flow (Figure 9) would be interesting.
3. Please clarify and justify the choice of parameterizations, including when CLEO is designed to switch between different parameterizations, and when a single parameterization has been fixed. For instance, condensation and ventilation appear to have a single fixed option, whereas coalescence leaves room for multiple implementations. Please explain and justify the mixing of various implementation components, such as describing the probabilities of coalescence/breakup/rebound from Testik while utilizing the coalescence efficiency of Straub 2010. It would further be useful to understand where CLEO's chosen parameterizations differ from those adopted by other production-ready SDMs (LibCloud, SCALE, PySDM). Differences between PySDM and CLEO in collision implementation are well-documented, but understanding any differences in CLEO's default choice for other dynamics such as the collision kernel and sedimentation velocity would be helpful to the modeling community.
4. Please describe some implementation details that are relevant to coupling with LES or another dynamical model:
- o What are the parameterized source and sink terms to Eulerian tracers including latent heating, moisture/buoyancy sources, or any momentum feedbacks during transport?
- o At a minimum, it would be helpful to describe the prognostic variable sets that are natively compatible with the current implementation of CLEO, and which would require interpolation and/or additional coupling intricacies. For instance, many large scale models operate in pressure coordinates rather than z; others have different choices of moisture and thermodynamic prognostic variables. In my experience, converting between these native

prognostic variables and those required by the SGS scheme can result in qualitative differences simply due to small differences in a choice of thermodynamic constants, so it is helpful to understand the native variable set that is most compatible with CLEO.

Minor Comments:

- The CLEO acronym is never defined
- L92-93: "Droplets with a radius... same terminal velocity" needs a reference or proof.
- L129: "references in Morrison et al. 2020" is not an appropriate citation; select the appropriate supporting references and cite them directly
- Please clarify the rationale for terminating a simulation rather than removing an empty superdroplet (L207)
- I could not find the citation for Bayley 2025b; Bayley 2025a needs an updated URL with the relevant DOI
- L228: the result of collision is only deterministic in certain cases in CLEO, but is still probabilistic in that it uses the random number (Monte Carlo) in other cases
- Can you clarify for a non-computer scientist what a C++20 concept is?
- L358 and elsewhere: clarify that adaptive sub-time-stepping is only implemented in condensation, not in coalescence (whereas PySDM v2 includes a coalescence adaptive time step)
- The lightest yellow in Figure 7 is difficult to see and does not print well.
- A table of notation would be helpful for reference

Typos:

- L32: "effect" → affect
- L110: "upto"
- L147: "we prescribe for the collision probability" – a leftover/hanging phrase?
- L312: "was" → were

---

## Referee Comment (RC2)

Bayley et al. provides a description of the parameterizations implemented in their super-droplet method model CLEO. As a model description paper, the manuscript provides valuable information in terms of the implementation such that one could examine the choices of the CLEO model and determine that they are correctly implemented. Additionally, such details are essential for understanding how CLEO could be used within other models and/or compared against other models. This manuscript presents a valuable addition as it presents a novel approach for microphysics, discusses the implementation details of said microphysics and particle motion due to mean flow. Such a framework is important for addressing the shortcomings of simplified cloud microphysics representation. Therefore, I recommend this manuscript should be suitable for publication pending revisions. I request that authors address comments and consider possible suggestions that may clarify and strengthen the manuscript further.

**1 Major comments**

- The general use of term Eulerian feels potentially confusing as a reader given the paper discusses Eulerian in both the sense of space and also in the sense of sectional microphysics. The presented model relies on a Eulerian model for solving wind field (and water vapor) but is Lagrangian in both particle position and in terms of microphysics.

- The authors could briefly discuss how CLEO can interface with other models, especially since it is presented here in a simplified 2D setup. Even if the companion paper covers this in detail, a short explanation would help readers understand its applicability, particularly regarding particle motion (see Section 4).

- In general, it is not really clear to me what CLEO is and is not. It is not 100% clear how it relates to the original SDM or PySDM, if it is merely an extension of a model or brand new model.

- Section 4 does not include any stochastic component for turbulence. While this is an intentional choice, if may be unclear to readers why this decision was made, whether or not there are justifications to this or if this is a possibly a typical assumption in these types of models.

- Beyond the noted parameterizations and particle motion, were there other significant assumptions in the model process design? Listing these briefly would help potential users or those comparing CLEO with their own models understand it's key assumptions.

- Section 5 is entitled "Validations" while it appears to contain, at least mostly, model verification. The authors should clarify whether or not certain aspects are for model verification or model validation. Dividing up this section of four tests into subsections may also be of benefit, such as one section for three individual processes and another section that puts everything together.

- I suggest that the authors find a way to possibly present the convergence of the particle field of Figure 9 in a more meaningful way rather than relying on visual inspection.

- This paper has been submitted as a Model description paper. I would suggest that the following is considered:

  - The scope and limitations of the approach adopted for CLEO could be expanded as the authors deem applicable. The manuscript may benefit from a paragraph or brief subsection regarding a discussion of scope of applicability and limitations. This is helpful for people to evaluate the applicability of CLEO and be aware of its weaknesses or aspects that are currently omitted. Example of which, how effects of sub-grid scale turbulence were noted to be neglected in Section 4.

**2 Minor comments**

- I believe that the title, per journal guidelines, should include information regarding the version number. This number appears to be v0.52.0 in the code availability section of this manuscript. However, this may be complicated by the fact that the companion paper regarding performance appears to have used a different version of CLEO.

- I believe Equations 12 and 13 should use $S_S$ and $S_L$ rather than $R_S$ and $R_L$, since the latter are not defined in the text. Please verify or correct accordingly. Also check the consistency of the defintion of $S_c$ since it uses $R_S$ and $R_L$, and it isn't clear why it would not just use $R_j$ and $R_k$.

- Line 334: "...and a log standard deviation of 1.4" appears to be incorrect. I believe this should be standard deviation and not the log of the standard deviation. Please verify and correct if necessary.

**3 Technical errors**

- Line 176: should $\sigma$ actually be $\sigma_l$?

- Figure 8 introduces $N$ as number of superdroplets when $n_s$ was introduced earlier.

- In the reference section, many of the DOIs appear to be mangled with repeated doi.org.